# Lower alpha, higher beta, and similar gamma diversity of saproxylic beetles in unmanaged compared to managed Norway spruce stands

**Oskar Gran** *

Department of Biological and Environmental Sciences, University of Gothenburg, Gothenburg, Sweden

* oskar.gran@bioenv.gu.se

## Abstract

Strong anthropogenic pressures on global forests necessitate that managed forests be evaluated as habitat for biodiversity. The complex pattern of habitat types created in forestry systems is ideal for analyses through the theoretical framework of alpha (local), gamma (total) and beta (compositional) diversity. Here I use saproxylic beetles, a species-rich threatened group, to compare four Norway spruce-dominated habitats representative of the boreal forest landscape of northern Europe: unmanaged semi-natural stands, nature reserves, unthinned middle-aged production stands and commercially thinned production stands. The beetles (in total 38 085 individuals of 312 species), including red-listed ones and three feeding guilds (wood consumers, fungivores and predators) were studied in 53 stands in central-southern Sweden, in two regions with differing amounts of conservation forest. Alpha diversity of saproxylic, but not red-listed, beetles was higher in the thinned stands than in the semi-natural stands, and did not differ for the other forest types. Beta diversity of saproxylic beetles was higher in unmanaged semi-natural stands than in the other forest types, but species composition did not differ noticeably. Furthermore, red-listed saproxylic beetles had higher gamma diversity in unmanaged semi-natural stands in the region with more conservation forest, but not in the one with less such forest. The local factors dead wood volume and dead wood diversity did not influence alpha diversity of beetles, but increasing canopy openness had a minor negative influence on saproxylic and red-listed beetles. While the local scale (alpha diversity) indicates the potential for managed forests to house many saproxylic beetle species associated with spruce forests in this boreal landscape, the larger scales (beta and gamma diversity) indicate the value of unmanaged forests for the conservation of the entire saproxylic beetle fauna. These results show the importance of analyses at multiple levels of diversity (alpha, beta, gamma) for identifying patterns relevant to conservation.

## Introduction

Globally, forest biodiversity faces many anthropogenic pressures, with European forests among the most affected [1]. Conservationists have emphasized the role of modern forestry

**Data Availability Statement:** All data are archived and available via the Open Science Framework: https://doi.org/10.17605/OSF.IO/2EC68.

**Funding:** This study was funded by 'Stiftelsen Oscar och Lili Lamms Minne' (DO2016-0005, [[[http://stiftelsenlamm.a.se)]http://stiftelsenlamm.

a.se]http://stiftelsenlamm.a.se)]http://
stiftelsenlamm.a.se]http://stiftelsenlamm.a.se)]
http://stiftelsenlamm.a.se]http://stiftelsenlamm.a.
se)]http://stiftelsenlamm.a.se), 'Carl Tryggers
Stiftelse för Vetenskaplig Forskning' (CTS16:171,
[[[https://www.carltryggersstiftelse.se)]https://
www.carltryggersstiftelse.se]https://www.
carltryggersstiftelse.se]https://www.
carltryggersstiftelse.se)]https://www.
carltryggersstiftelse.se]https://www.
carltryggersstiftelse.se)]https://www.
carltryggersstiftelse.se), 'Helge Ax:son Johnsons
stiftelse' (F20-0184, [[[https://www.haxsonj.se)]
https://www.haxsonj.se]https://www.haxsonj.se)]
https://www.haxsonj.se]https://www.haxsonj.se)]
https://www.haxsonj.se]https://www.haxsonj.se)]
https://www.haxsonj.se) and 'Herbert och Karin
Jacobssons Stiftelse' (11/v20, [[[https://
hkjacobssonstiftelsen.se)]https://
hkjacobssonstiftelsen.se]https://
hkjacobssonstiftelsen.se)]https://
hkjacobssonstiftelsen.se]https://
hkjacobssonstiftelsen.se)]https://
hkjacobssonstiftelsen.se]https://
hkjacobssonstiftelsen.se)]https://
hkjacobssonstiftelsen.se). The funders played no
role in the study design, data collection, analysis,
publication or preparation of the manuscript.

**Competing interests:** The authors have declared
that no competing interests exist.

practices, especially clearcutting, in driving homogenization and extinctions among forest species [2–7]. While more protected forest is needed, there is increasing recognition of the need for conservation measures also within the managed forest matrix [8–11] in order to reach goals of sustainable forest management (e.g. the Convention on Biological Diversity, UN Sustainable Development Goal 15, and the EU Biodiversity Strategy for 2030). It is necessary to identify taxa that are especially sensitive to forestry operations, and other taxa that may be maintained within managed forests. In this study, I compare species diversity and composition of saproxylic (wood-living) beetles between two managed and two unmanaged Norway spruce (*Picea abies* (L.) H. Karst.) forest types, to clarify how a diverse species group of conservation concern is distributed in and affected by the managed forest matrix in boreal central-southern Sweden.

Saproxylic insects are ecologically important and diverse, and beetles are the most diverse saproxylic insects [12]. In Europe, 18% of assessed saproxylic beetle species are classified as threatened (red-list classes VU, EN, CR), with logging singled out as a primary cause of species declines [13]. In Sweden, 400 out of 1153 (35%) saproxylic beetle species are red-listed (18% classified as threatened, [14]).

The fauna of saproxylic beetles may differ not only between managed and unmanaged forests, but also between management stages [15]. In a previous study we found as many saproxylic beetle species overall, but fewer red-listed species, in pre-commercially thinned young spruce stands compared to unmanaged stands [16]. In many countries, thinning (pre-commercial and commercial) is done on much larger areas each year than final felling—in Sweden nearly three times larger [17] (see [16] for further examples). Young to middle-aged forestry stands are now much more common in Fennoscandian landscapes than before industrialization [18], but are often overlooked in conservation research. At the same time, the long-term effect of thinning on saproxylic beetles is unclear [19,20], and middle-aged stands that have not been recently thinned are also an important part of the forestry landscape.

In the present study I use recently commercially thinned spruce-dominated stands ("thinned stands" below) and spruce-dominated stands without recent forestry intervention ("unthinned stands" below). As a semi-natural reference I use Woodland Key Habitats ("WKHs" below); small, semi-natural forest stands, identified by their biodiversity values and scattered throughout the managed forest landscape in northern Europe [21]. WKHs are important in regional forest conservation [22–25], but to complement them, I use spruce-dominated nature reserves as larger natural forests ("reserves" below). Given the importance of geographic context to biodiversity patterns, and in light of earlier results showing the importance of the amount of WKH in the surrounding landscape to red-listed saproxylic beetles associated with oak [26], I use sites from two large regions that differ in the concentration of WKHs. Contrasting results between these two regions could indicate the importance of the surrounding landscape in determining these patterns.

I use a framework of diversity divided into alpha-, beta- and gamma diversities [27,28], where alpha diversity represents the local diversity of a single stand, beta diversity represents the degree of variation in community composition among stands within a forest type, and gamma diversity represents the total diversity of all stands of one type in the region studied. Many studies of diversity differences between managed and unmanaged forests deal only with the average diversity of individual stands (i.e. alpha diversity). However, species diversity patterns are highly scale dependent [28,29], and patterns at the local scale are often different or reversed at larger scales [30,31]. The framework of alpha, beta, and gamma diversity can give a more complete view, and reveal patterns that would otherwise be missed (e.g. [32–34]). Forestry may increase, decrease or leave unaffected alpha, beta and gamma diversity separately [35], and failing to properly consider the scale of diversity patterns may lead to e.g. poor

management recommendations [36]. For example, a higher degree of habitat heterogeneity among unmanaged than managed forests [4,5] could mean that clear differences in species diversity are seen only at the beta or gamma scale.

The amount and diversity of dead wood in a forest seem to be major determinants of saproxylic beetle diversity [37]. Canopy openness is also important, with many species preferring sun-exposed dead wood [38] while others are associated with more shaded wood [39]. These local environmental factors are highly affected by forestry operations. Dead wood recruitment in managed forests follows thinning and felling operations, creating temporary pulses of dead wood [40–42]. On average, dead wood volume and diversity are lower in managed than in unmanaged conifer forests in northern Europe [3,43]. Canopy openness in even-aged forestry is cyclical, with open conditions after clear-cutting and increasingly closed conditions as stands age, counteracted by thinnings. In unmanaged forests, dead wood recruitment and canopy openness is governed by natural succession, disturbances, and 'gap dynamics' [2,44]. This study consequently also examines the influence of these local environmental variables on beetle diversity.

Because of the high ecological diversity of saproxylic organisms, it is useful to divide them into functional groups which may respond differently to environmental factors [45,46]. Here I study saproxylic beetles as a whole ("saproxylic beetles" below) and divide them into three feeding guilds (wood consumers, fungivores and predators) and red-listed beetles.

The study aims to test the following specific predictions:

1. The alpha diversity of saproxylic beetles does not differ between thinned spruce production stands, spruce-dominated Woodland Key Habitats and reserves, as there are many species adapted to each forest type. Unthinned spruce production stands, with less dead wood and canopy openness, have lower alpha diversity of saproxylic beetles than the other forest types.

2. The alpha diversity of red-listed beetles is higher in unmanaged than in managed stands, owing to a higher concentration of rare and valuable dead wood substrates. Unthinned stands have lower alpha diversity than the other forest types.

3. Certain species are better adapted to unmanaged stands than managed stands and vice versa, consequently species composition differs between managed and unmanaged forest types.

4. The alpha diversity of saproxylic and red-listed beetles increases with the amount and diversity of dead wood, and with canopy openness.

5. The beta diversity of saproxylic beetles is higher in unmanaged than in managed stands due to higher among-habitat heterogeneity of substrates and structures, especially dead wood.

6. The gamma diversity of saproxylic beetles is higher in unmanaged than in managed stands due to higher habitat heterogeneity. Unthinned stands have lower gamma diversity than the other stand types.

7. The gamma diversity of red-listed beetles is higher in unmanaged than in managed stands, and the difference is larger in the region with a higher concentration of WKHs. Unthinned stands have lower gamma diversity than the other stand types.

8. The gamma diversity of fungivores is highest in WKHs and reserves because of higher dead wood and fungal diversity; wood consumers are most diverse in thinned stands because of the pulse of newly dead wood; and predators are most diverse in the WKHs and reserves owing to a higher sensitivity to forestry; alternatively, predators are most diverse in the thinned stands because of high numbers and diversity of wood consumer prey.

## Methods

### Study regions and stands

The present study combines data from two different sampling years and regions in the hemi-boreal zone of southern/central Sweden; Jönköping (roughly corresponding to Jönköping county) in 2017 and Örebro (roughly corresponding to Örebro county) in 2018.

The 2017 Jönköping sample region consists of 10 sites, each with a thinned production stand paired in proximity to a Woodland Key Habitat (WKH) stand. In addition to these 10 pairs, I included three sites with (unpaired) nature reserve stands in the same region. The 2018 Örebro sample region consists of 10 sites, each with one thinned production stand, one unthinned production stand and one WKH in proximity. Consequently, the complete sample from both regions consists of 20 WKH stands, 20 thinned stands, 10 unthinned stands and 3 reserve stands (Fig 1).

Both regions are forested at around 70% of land area [17], and dominated by Norway spruce (~45–50% of tree volume) followed by Scots pine (*Pinus sylvestris* L., ~35%) and birches (*Betula*

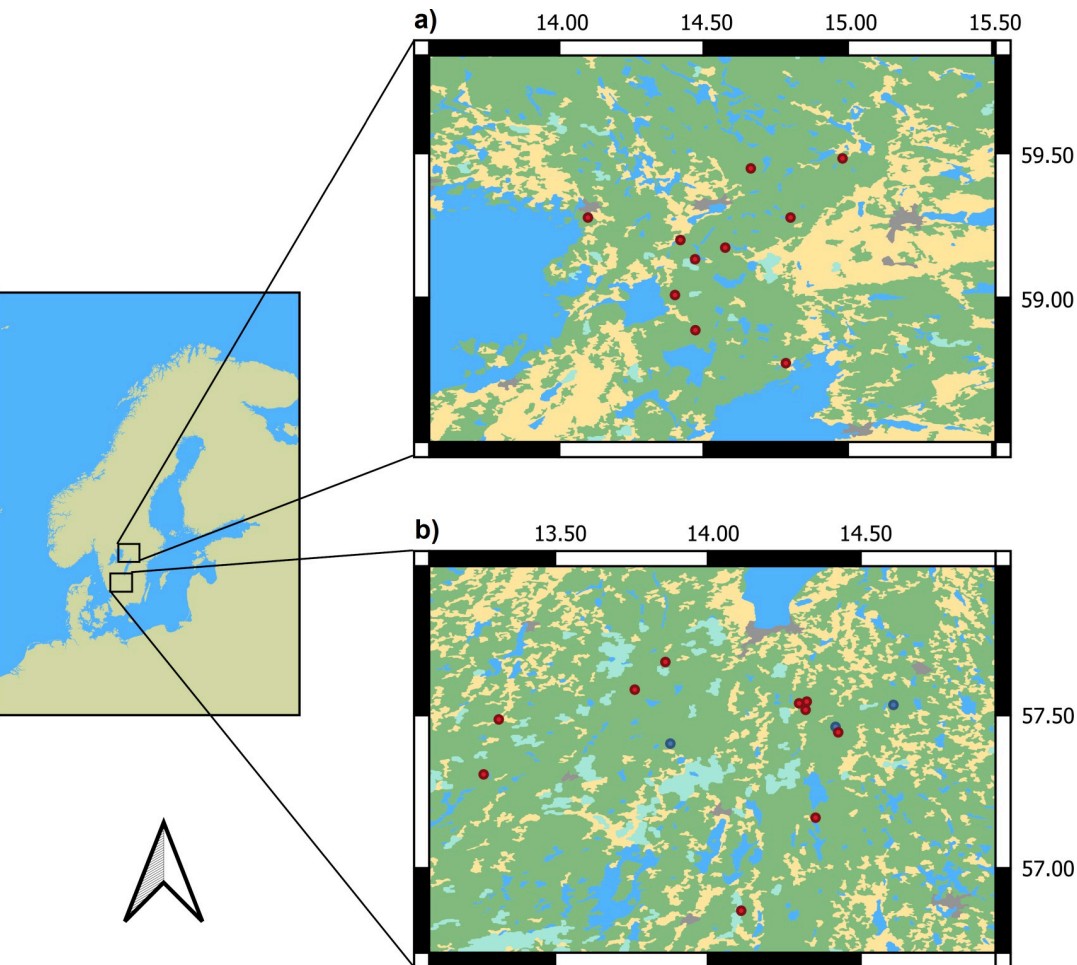

**Fig 1. Map of study sites.** a) Study sites in the northern region, corresponding roughly to Örebro county. Each red dot represents a study site consisting of one thinned production stand, one unthinned production stand and one Woodland Key Habitat stand. b) Study sites in the southern region, corresponding roughly to Jönköping county. Each red dot represents a study site consisting of one thinned production stand and one Woodland Key Habitat stand. Blue dots represent unpaired nature reserve stands. Green is forested land, teal is wetland, yellow is open (agricultural) land, and grey is urban land. Coordinates are in WGS84, with decimal degrees as units. Background map provided by [47].

*pendula* Roth/*Betula pubescens* Ehrh., ~11%) [48]. Although the two regions have many similarities, including a long history of human forest use [49,50], Örebro county has a markedly higher percentage of forest area that is strictly protected; 4.5% vs 1.8% in Jönköping county [51]. The number and total area of WKHs is also markedly higher in Örebro than in the Jönköping region (defined here by a rectangle encompassing all study sites in each region): ~4.4 WKHs/1000 ha, ~1.4% of total area in Örebro vs ~2.6 WKHs/1000 ha, ~0.5% of total area in Jönköping [52].

Mean yearly temperature in the two regions was 6°C and mean yearly precipitation was between 700–1000 mm during the current standard period of 1991–2020. During the sampling period in 2017 (May-July), the mean temperature in the Jönköping study region was 11–15°C and mean monthly precipitation 20–125 mm. During the sampling period in 2018 (May-July, a hot and dry summer), the mean temperature in the Örebro study region was 14–21°C, and mean monthly precipitation 25–50 mm [53]. Jönköping sites are on average 237, and Örebro sites on average 153 meters above sea level.

All sampled sites except the natures reserves are owned by the state forest company Sveaskog. The thinned stands were recently commercially thinned Norway spruce production stands. The unthinned stands were middle-aged Norway spruce production stands without recent forestry interventions. The WKHs were older Norway spruce-dominated stands. The following criteria were used in selecting sites from the Sveaskog database: forest type (WKH or managed), stand age (over 35 for managed forests), time since intervention (more than 10 years for unthinned stands, 1–5 years for thinned stands), size (above 0.5 ha for WKHs), distance between paired stands (within 2.5 km), tree species (at least 70% Norway spruce). The three reserves in Jönköping were selected as the largest in the region dominated by old Norway spruce forest. All sites were embedded in a forestry mosaic dominated by Norway spruce stands, typical of central-southern Sweden. Stand characteristics per forest type and region are given in Table 1.

## Measurement of local environmental factors

I sampled beetles in each stand using two insect traps, and recorded environmental variables around traps. In a 5.5 m radius from each trap, I measured living stems 2 m and taller, for each stem recording tree species, height and diameter at breast height (1.3 m). I measured the diameter of multi-stemmed hazel (*Corylus avellana* L.) at the ground (not individual stems). Fig 2 shows tree composition for each forest type. Norway spruce dominated all sites, followed by birches (*Betula pubescens*/*B. pendula*, similar taxa, pooled in this study) and Scots pine. WKHs had the greatest diversity of tree species.

I sampled all dead wood objects over 1 cm in diameter within a 10 m radius around each trap. I recorded tree species, diameter at both ends, height/length, type, and decay stage. In cases of large piles of small objects (diameter 1–3 cm), I measured representative objects and extrapolated to the full count. Dead wood type was classified in 4 categories: logs & branches (lying objects), stumps (height < 1 m), snags (height > 1 m) and dead trees (branches in crown more or less intact). For some snags where the top was too high to reach, diameter at the top was estimated visually. I classified wood decay in 4 stages: 1) newly dead wood (around 1 year or younger), leaves and/or small twigs still attached, bark intact, 2) older than one year but wood still hard, bark still mostly intact but looser, 3) older wood, partly soft, much bark gone, 4) old, heavily decayed and deformed wood.

I calculated dead wood diversity around each trap as the number of unique combinations of 4 different factors with a varying number of levels, similar to the method used in [54]: tree species (12 species), type (4 types), decay stage (4 stages) and diameter class (3 classes). For diameter class, I used the average of the diameters at the two ends, and used the classes 1–10 cm, 10–30 cm, 30+ cm from [55].

**Table 1. Stand characteristics.** Descriptive statistics for environmental factors and stand characteristics per forest type and region. The number of stands of each forest type per region is given in parentheses. WKH = Woodland Key Habitat.

| Region | Forest type | Dead wood volume (m³/ha) | | | Dead wood diversity | | |
|---|---|---|---|---|---|---|---|
| | | *min* | *max* | *mean (±SD)* | *min* | *max* | *mean (±SD)* |
| Jönköping | WKH (10) | 7.3 | 214.4 | 72.6 (±50.2) | 3 | 24 | 12.6 (±4.9) |
| Jönköping | Thinned (10) | 4.7 | 36.7 | 15.2 (±8.3) | 9 | 17 | 12.1 (±2.6) |
| Jönköping | Reserve (3) | 27.5 | 89.5 | 57.7 (±26.1) | 6 | 13 | 9.3 (±2.3) |
| Örebro | WKH (10) | 6.7 | 267.4 | 92.2 (±74.2) | 6 | 20 | 11.6 (±3.8) |
| Örebro | Thinned (10) | 4.5 | 67.0 | 17.3 (±15.5) | 7 | 21 | 12.8 (±3.8) |
| Örebro | Unthinned (10) | 0.7 | 41.8 | 13.0 (±11.2) | 5 | 21 | 10.3 (±4.2) |
| **Region** | **Forest type** | **Stand age (years)** | | | **Stand size (ha)** | | |
| | | *min* | *max* | *mean (±SD)* | *min* | *max* | *mean (±SD)* |
| Jönköping | WKH (10) | 55 | 149 | 111 (±26) | 0.9 | 7.8 | 3.1 (±2.5) |
| Jönköping | Thinned (10) | 37 | 51 | 43 (±5) | 0.9 | 31.7 | 7.6 (±9.4) |
| Jönköping | Reserve (3) | | | | 70.3 | 284.5 | 187.6 (±108.6) |
| Örebro | WKH (10) | 81 | 161 | 114 (±28) | 0.9 | 20.7 | 5.4 (±6.0) |
| Örebro | Thinned (10) | 35 | 50 | 41 (±6) | 1.1 | 17.8 | 6.9 (±6.1) |
| Örebro | Unthinned (10) | 35 | 68 | 49 (±11) | 1.0 | 6.8 | 3.4 (±2.0) |
| **Region** | **Forest type** | **Basal area (m²/ha)** | | | **Average living tree height (m)** | | |
| | | *min* | *max* | *mean (±SD)* | *min* | *max* | *mean (±SD)* |
| Jönköping | WKH (10) | 19.1 | 137.1 | 59.8 (±24.1) | 8.9 | 22.5 | 16.2 (±4.6) |
| Jönköping | Thinned (10) | 19.6 | 58.2 | 32.3 (±9.6) | 10.8 | 19.7 | 14.5 (±2.6) |
| Jönköping | Reserve (3) | 34.9 | 76.9 | 54.4 (±16.5) | 12.3 | 23.2 | 16.9 (±4.5) |
| Örebro | WKH (10) | 17.2 | 136.4 | 57.1 (±29.4) | 7.9 | 29.9 | 17.4 (±6.3) |
| Örebro | Thinned (10) | 13.4 | 50.5 | 34.1 (±9.4) | 11.4 | 19.3 | 15 (±2.3) |
| Örebro | Unthinned (10) | 23.1 | 70.4 | 49.7 (±14.0) | 8.4 | 23.4 | 15.8 (±4.3) |
| **Region** | **Forest type** | **Canopy openness (%)** | | | **Time since thinning** | | |
| | | *min* | *max* | *mean (±SD)* | *min* | *max* | *mean (±SD)* |
| Jönköping | WKH (10) | 19 | 41 | 27 (±6) | | | |
| Jönköping | Thinned (10) | 19 | 53 | 34 (±7) | 1 | 5 | 2.9 (±1.4) |
| Jönköping | Reserve (3) | 24 | 37 | 30 (±5) | | | |
| Örebro | WKH (10) | 14 | 48 | 24 (±9) | | | |
| Örebro | Thinned (10) | 16 | 38 | 24 (±6) | 1 | 5 | 3.3 (±1.6) |
| Örebro | Unthinned (10) | 17 | 30 | 22 (±4) | | | |

I calculated the volume of each dead wood object using the formula for a conical frustum: $V = \pi h/3 \times (R^2 + R \times r + r^2)$, where h is length, and R and r are the radiuses at each end. For dead trees, I used volume functions intended for the specific tree species [56]. Finally, I pooled the volume of all objects around each trap to obtain m³/ha. Fig 3 shows characteristics for all 2896 recorded dead wood objects, which were dominated by Norway spruce in all forest types, followed by birches.

I estimated canopy openness from photos taken from the ground straight up at both sides of each trap. The images were processed with a high-contrast, greyscale filter, then analyzed for "mean grey value" in ImageJ 1.50b, to estimate percentage canopy openness. Environmental variables per forest type and region are summarized in Table 1.

## Sampling of beetles and handling of species data

I sampled beetles using IBL-2 flight interception traps (CHEMIPAN, Warsaw), consisting of 0.3 m² triangular plastic sheets suspended between a white plastic roof and two gutters, with a

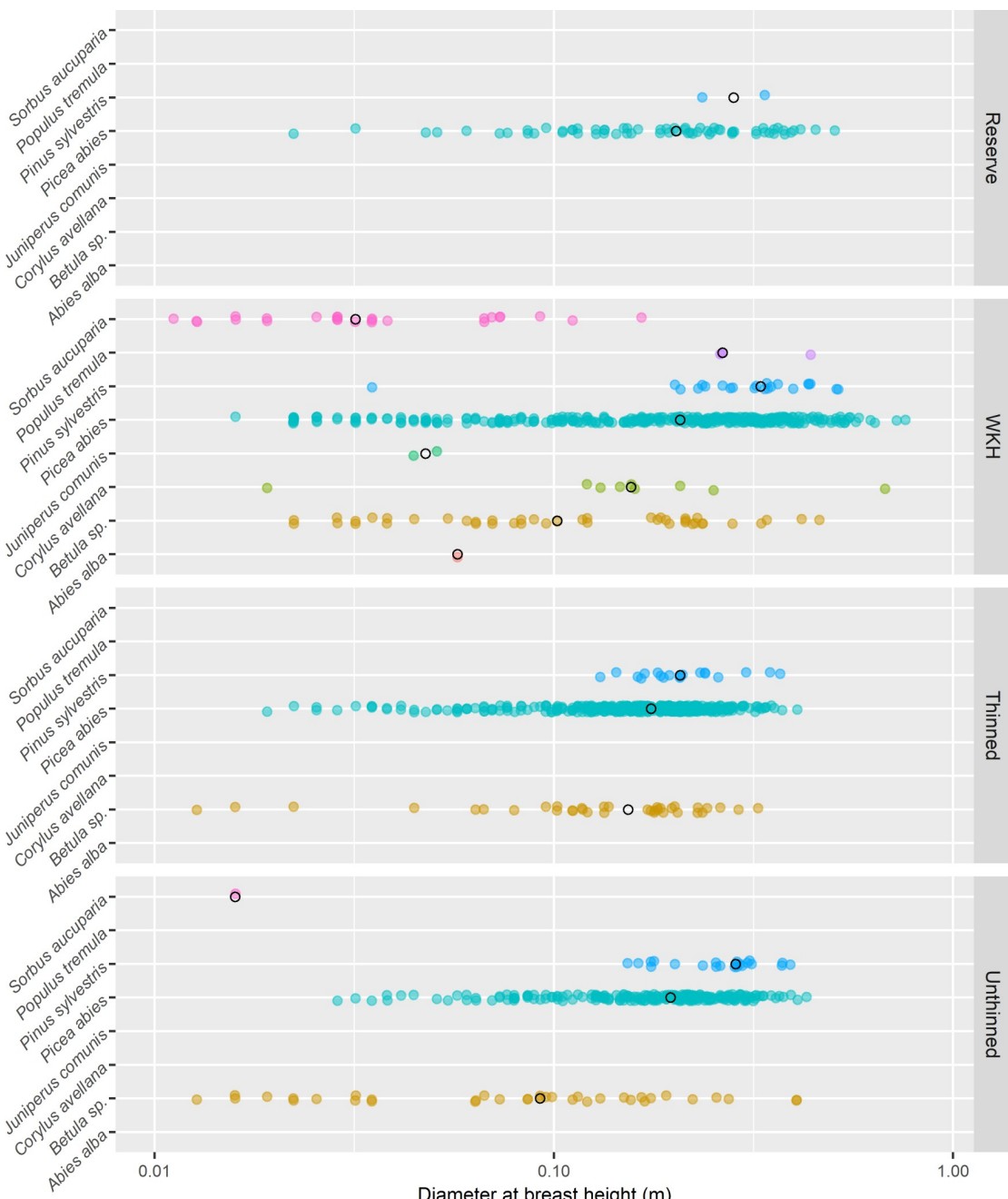

**Fig 2. Living tree measurements.** Stems measured within 5.5 m radius around each trap, summed for each forest type. Black circles represent median values. *Corylus avellana* was measured at base, not breast height. Note the logarithmic scale on the x-axis. A slight jitter has been applied to increase readability of overlapping points. Sample sizes are not equal for the four forest types (reserves: 3, Woodland Key Habitats (WKHs): 20, thinned stands: 20, unthinned stands: 10).

collection jar at the bottom. These were hung between two living Norway spruces, at breast height. I used two traps per stand, about 25 m apart. In both years, traps were set up early May, emptied once in mid-June, and once at the end of July. Permission to sample stands outside of protected areas was given by Sveaskog, and no further permits were required under Swedish law. Permission to sample the three nature reserves was given by the county administrative board of Jönköping, permit number 521-2288-17.

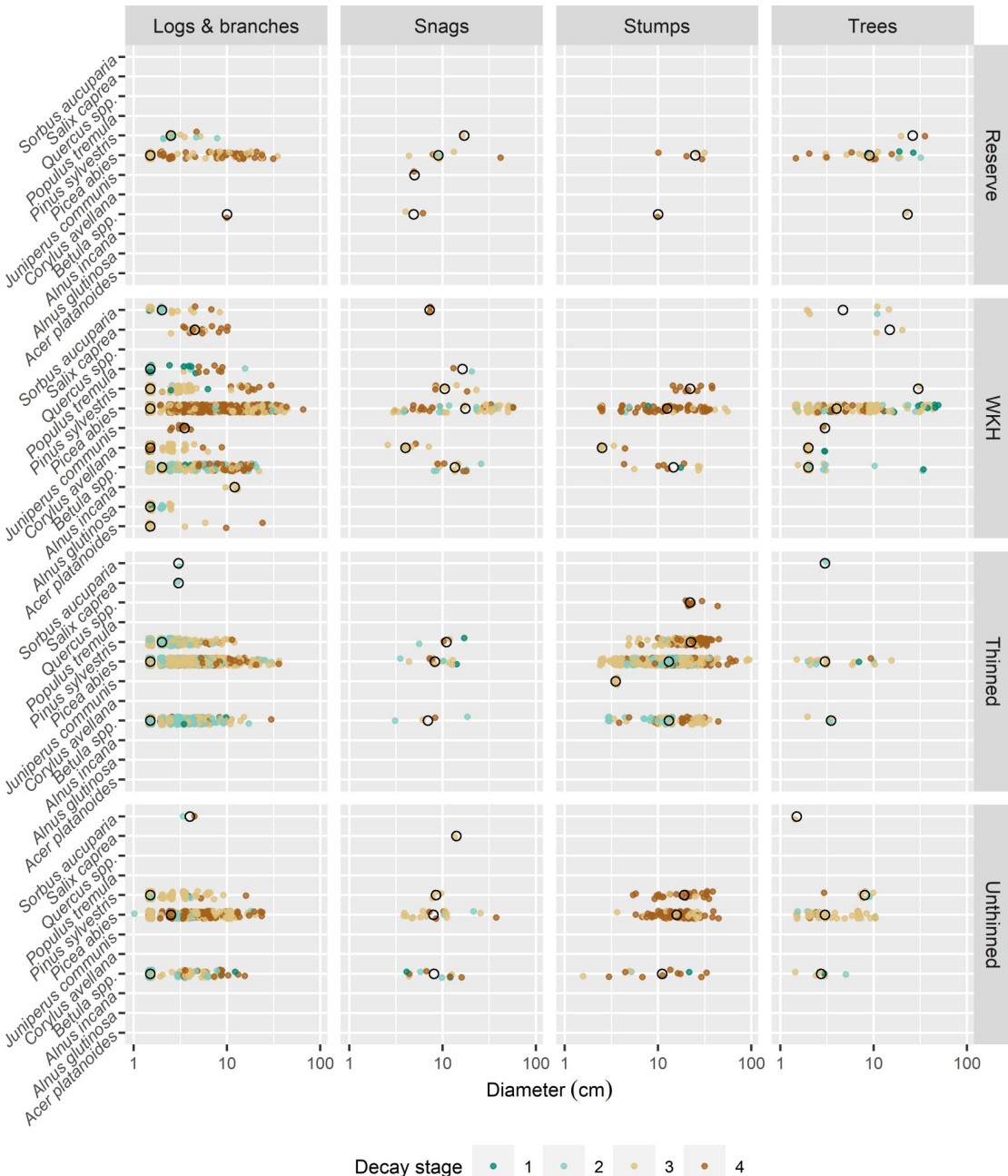

**Fig 3. Dead wood measurements.** Dead wood objects measured in a 10 m radius around each trap. Black circles represent median values. Decay stage 1 is newly dead wood, 2 and 3 are intermediate decay, and 4 is heavily decayed. Diameter is the average of the diameter at the two ends. Note the logarithmic scale on the x-axis. A slight jitter has been applied to increase readability of overlapping points and draw order has been randomized. Sample sizes are not equal for the four forest types (reserves: 3, Woodland Key Habitats (WKHs): 20, thinned stands: 20, unthinned stands: 10).

All saproxylic beetles were identified to species level based on morphological characters. The families Staphylinidae and Ptilidae were excluded due to difficulty in identification, except for subfamilies Pselaphinae, Scydmaeninae and Scaphidiinae. I classified saproxylic beetles into feeding guilds (wood consumers, fungivores, predators) based on [55,57]. When appropriate, species were classified in more than one feeding guild. I further classified red-listed

beetles, based on inclusion in any of the Swedish red-lists since 2000 [14,58–61]. Using older red-lists allowed for a bigger pool of species, as red-listed species are by their very nature often difficult to sample [62]. Previously red-listed species are generally still of conservation interest.

Trap-level data were used for all analyses except for species accumulation curves, where I used stand level data (two traps per stand pooled) to avoid pseudoreplication in that method. The sample consists of 40 traps in WKHs, 40 traps in thinned stands, 20 traps in unthinned stands and 6 traps in reserves.

## Statistical analyses

I used generalized linear mixed models (GLMMs) to test the influence of forest type and local environmental factors on the average per-trap number of saproxylic and red-listed species (i.e. alpha diversity). Model 1 used number of saproxylic species per trap as the response variable, and model 2 used the number of red-listed species per trap, with the other factors being the same in both models. Forest type (WKH, reserve, thinned stand, unthinned stand), Region (Jönköping, Örebro) and the three continuous environmental variables (Deadwood volume, Deadwood diversity, Canopy openness) were fixed factors. All continuous variables were standardized before analysis. WKHs and Jönköping were reference levels for the two categorical variables, meaning that effect estimates in the results for the other levels represent deviations from these. I included random intercepts per Site, and Stand nested within Site. I ran the models in R 4.1.0 [63] using the R package glmmTMB [64], with a Poisson distribution and log link function. I inspected residual plots using the DHARMa R package [65], and checked for overdispersion by comparing the sum of squared Pearson residuals to the residual degrees of freedom [66], finding no issues in either case. Statistical significance in the GLMM models was assessed using profile likelihood confidence intervals calculated with glmmTMB, as these are generally more informative and less prone to misinterpretation than p-values [67,68].

I compared gamma diversity of each category (saproxylic beetles, red-listed beetles, the three feeding guilds) among the forest types using species accumulation curves and conservative 95% confidence intervals constructed in EstimateS v.9.1.0 [69]. I extrapolated each sample to twice the sample size using the Chao2 asymptotic estimator [70]. To obtain a measure of evenness for each forest type, I constructed rank-abundance curves per stand type and region.

I tested differences in beta diversity of saproxylic beetles among the four forest types using PERMDISP, with p-values calculated based on 999 permutations using PRIMER v.7.0.13 [71] and the PERMANOVA+ add-on [72]. P-values were used as confidence intervals are not available for this analysis. The choice of dissimilarity measure can have a drastic effect on results [73], so I used both the modified Gower dissimilarity measure with log base 10 [74] which takes into account species abundances, and the Sørensen dissimilarity measure, which does not.

Given a significant result in the PERMDISP analysis, I did not perform a PERMANOVA to test for differences in community composition as it cannot tell apart multivariate dispersion (i.e. beta diversity) and location (i.e. differences in community composition) [75]. Instead, I visualized species assemblages using NMDS, based on the modified Gower dissimilarity measure. In the NMDS, lack of overlap between sample points from different forest types would indicate differences in community composition, and forest types with less clustered points would indicate higher beta diversity. The stress in the two-dimensional NMDS was slightly above the recommended limit of 0.2 (0.24). Analyses of beta diversity and community composition were done only for saproxylic beetles, as red-listed beetles were too few for meaningful analysis. All plots were drawn using the 'ggplot2' R package [76].

**Table 2. Summary species data.** Species (spec.) and individuals (ind.) per species group and forest type. The number of stands per forest type is given in parentheses. WKH = Woodland Key Habitat.

| | Thinned (20) | | Unthinned (10) | | WKH (20) | | Reserve (3) | | Total (53) | |
|---|---|---|---|---|---|---|---|---|---|---|
| | Spec. | Ind. | Spec. | Ind. | Spec. | Ind. | Spec. | Ind. | Spec. | Ind. |
| **Red-listed species** | 29 | 226 | 22 | 145 | 27 | 228 | 6 | 20 | 37 | 619 |
| **Wood consumers** | 68 | 7211 | 56 | 4473 | 64 | 8342 | 26 | 1132 | 79 | 21 158 |
| **Fungivores** | 141 | 4418 | 115 | 2347 | 146 | 4594 | 69 | 502 | 171 | 11 861 |
| **Predators** | 54 | 2082 | 44 | 905 | 52 | 2014 | 22 | 308 | 63 | 5309 |
| **All saproxylics** | 260 | 13 480 | 215 | 7699 | 262 | 14 944 | 118 | 1962 | 312 | 38 085 |

## Results

The total sample consisted of 38 085 saproxylic beetle individuals of 312 species (18 283 individuals, 248 species in Jönköping in 2017; 19 802 individuals, 273 species in Örebro in 2018), representing 45 beetle families. For three of the forest types (commercially thinned stands, unthinned stands and Woodland Key Habitats) the proportion of species associated with Norway spruce was around 50–60% while the proportion associated with broadleaved trees was around 50% (with some species associated with both, and around 10% associated with Scots pine). For the nature reserves, the proportion associated with spruce was higher (around 70%) and the proportion associated with broadleaves lower (around 40%). Summary species data for the four forest types is given in Table 2.

### Alpha diversity and environmental factors

The GLMM for saproxylic beetles indicated a higher alpha diversity in thinned stands than in WKHs. Both of the managed stand types had effect estimates higher than the WKH reference level, although only thinned stands had confidence intervals not overlapping 1, indicating statistical significance (an estimated ~12% more species per trap, Table 3). For the reserves, the estimate was lower than the WKHs, although with overlapping confidence intervals. Region,

**Table 3. Saproxylic alpha diversity and environmental factors.** GLMM results for all saproxylic species. For the categorical predictors "Forest type" and "Region", Woodland Key Habitat (WKH) and Jönköping are the respective reference levels. The continuous predictors (Dead wood volume, Dead wood diversity, Canopy openness) have been unstandardized and all values have been back-transformed, giving odds ratios for the fixed effect estimates. The random factor "Stand" is nested within "Site". Confidence intervals (CI) not overlapping 1 (indicating statistical significance) have been marked in bold.

| Fixed effects | | | | |
|---|---|---|---|---|
| **Factor** | | **Estimate** | **CI 2.5%** | **CI 97.5%** |
| **Intercept** | | 55.00 | 49.26 | 61.41 |
| **Forest type (Reserves)** | | 0.94 | 0.74 | 1.18 |
| **Forest type (Thinned stands)** | | 1.12 | **1.01** | **1.24** |
| **Forest type (Unthinned stands)** | | 1.07 | 0.95 | 1.21 |
| **Region (Örebro)** | | 1.02 | 0.88 | 1.18 |
| **Dead wood volume (m³/ha)** | | 1.00 | 0.99 | 1.09 |
| **Dead wood diversity** | | 1.00 | 0.98 | 1.05 |
| **Canopy openness** | | 0.99 | **0.92** | **0.99** |
| Random effects | | | | |
| Factor | SD | | | |
| **Stand:Site** | 1.09 | | | |
| **Site** | 1.13 | | | |

**Table 4. Red-listed alpha diversity and environmental factors.** GLMM results for red-listed species. For the categorical predictors "Forest type" and "Region", Woodland Key Habitat (WKH) and Jönköping are the respective reference levels. The continuous predictors (Dead wood volume, Dead wood diversity, Canopy openness) have been unstandardized and all values have been back-transformed, giving odds ratios for the fixed effect estimates. The random factor "Stand" is nested within "Site". Confidence intervals (CI) not overlapping 1 (indicating statistical significance) have been marked in bold.

| Fixed effects | | | | |
|---|---|---|---|---|
| Factor | | Estimate | CI 2.5% | CI 97.5% |
| Intercept | | 2.74 | 2.02 | 3.70 |
| Forest type (Reserves) | | 0.68 | 0.31 | 1.37 |
| Forest type (Thinned stands) | | 0.99 | 0.71 | 1.40 |
| Forest type (Unthinned stands) | | 0.98 | 0.67 | 1.44 |
| Region (Örebro) | | 1.28 | 0.88 | 1.88 |
| Dead wood volume (m³/ha) | | 1.00 | 0.88 | 1.19 |
| Dead wood diversity | | 1.00 | 0.87 | 1.12 |
| Canopy openness | | 0.98 | **0.73** | **0.99** |
| Random effects | | | | |
| Factor | SD | | | |
| Stand:Site | 1.00 | | | |
| Site | 1.30 | | | |

dead wood volume and dead wood diversity had small estimates, all of which with confidence intervals indicating a lack of statistical significance. Canopy openness had a small, statistically significant negative influence on saproxylic beetles (equal to ~1% fewer species per percentage point of canopy openness). The among-site random effect was larger than any of the fixed effect estimates, while the among-stand random effect was smaller than for sites, although still larger than all fixed effects except the estimate for thinned stands (Table 3).

For red-listed beetles, the effect estimates for all forest types was lower than for WKHs, although all had confidence intervals overlapping 1, indicating a lack of statistical significance (Table 4). The estimate for the Örebro region was considerably higher than for the Jönköping reference level, although with wide confidence intervals overlapping 1. Dead wood volume and dead wood diversity had small estimates, with confidence intervals overlapping 1. Also for red-listed species, canopy openness had a small, statistically significant negative influence (equal to ~2% fewer species per percentage point of canopy openness). The among-site random effect was larger than any of the fixed effects, while the among-stand random effect was very small (Table 4).

None of the rank-abundance curves were noticeably different (Figs 1, 2 and S1), indicating similar evenness in all of the forest types.

In summary, the results indicate that for saproxylic beetles as a whole, commercially thinned spruce production stands have moderately higher local (alpha) diversity than spruce-dominated Woodland Key Habitats, but I found no difference for red-listed species or the other forest types. Furthermore, the results indicate that for both saproxylic beetles as a whole and for red-listed beetles, canopy openness has a small but significant negative effect on alpha diversity.

## Beta diversity and community composition

In the PERMDISP analysis of saproxylic beetles based on modified Gower dissimilarity, WKHs had the highest multivariate dispersion (e.g. beta diversity), followed by thinned stands, unthinned stands, then reserves (Table 5). The differences were statistically significant except

**Table 5. Beta diversity.** Multivariate dispersion (mean distances from centroid) for traps of each forest type with associated standard error (SE), from PERMDISP analysis based on modified Gower dissimilarity of saproxylic species abundance data. P-values below 0.05 in bold. WKH = Woodland Key Habitat.

| | Mean | SE | Pairwise comparisons | | |
| --- | --- | --- | --- | --- | --- |
| | | | Reserve | Thinned | Unthinned |
| **WKH** | 0.67 | 0.007 | t = 4.4; p = **0.001** | t = 2.5; p = **0.015** | t = 2.8; p = **0.012** |
| **Reserve** | 0.55 | 0.010 | | t = 4.6; p = **0.005** | t = 2.7; p = 0.095 |
| **Thinned** | 0.64 | 0.009 | | | t = 1.3; p = 0.319 |
| **Unthinned** | 0.62 | 0.014 | | | |

unthinned stands with thinned stands and reserves (overall results: F(3, 102) = 9.5, p = 0.002; pairwise comparisons in Table 5). The results were not substantially different using the Sørensen dissimilarity measure instead (S1 Table). The NMDS further corroborated this pattern in multivariate dispersion, with WKH points more spread out than those for thinned stands. The plot did not indicate any differences in multivariate location (i.e. community composition) among the forest types, with groups of points largely overlapping (Fig 4). However, the two

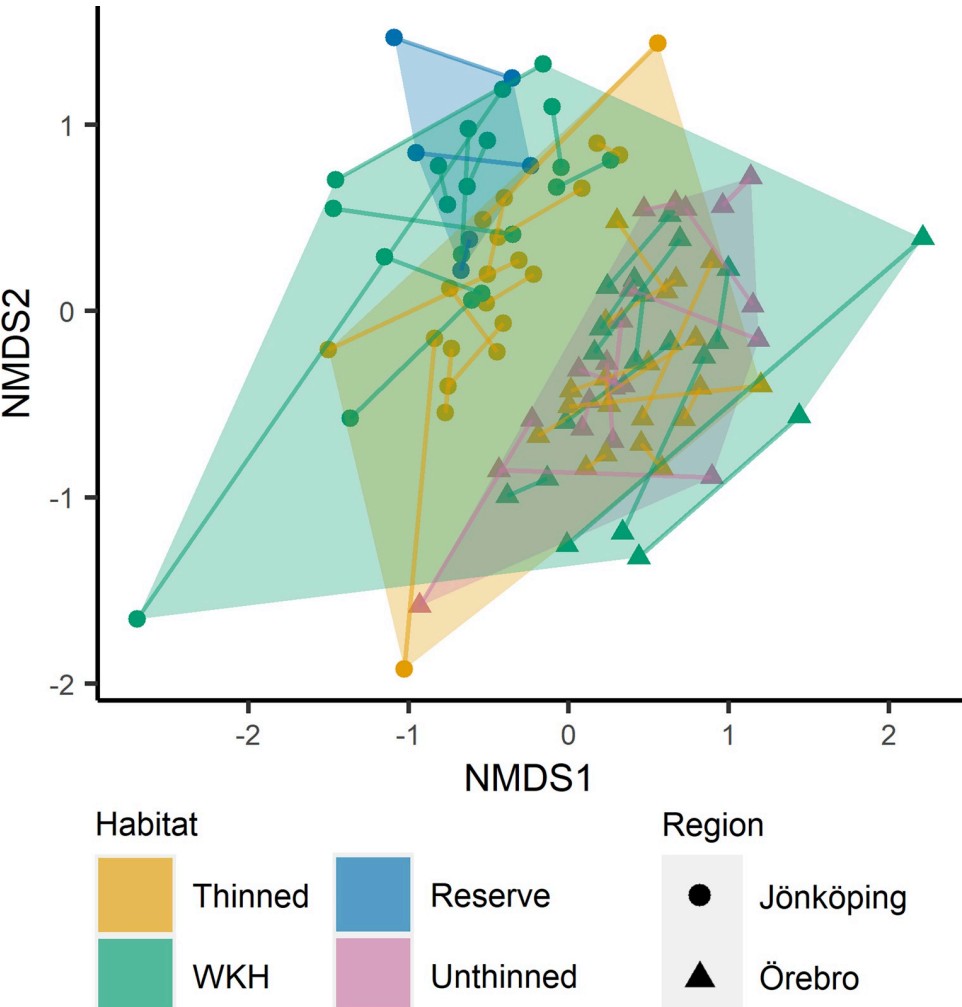

**Fig 4. Species assemblages.** NMDS of all traps, based on modified Gower dissimilarity of saproxylic species abundance data. Pairs of traps from the same stand have been connected with a line, and a hull has been drawn around all traps from the same forest type. K = 2, stress = 0.24. WKH = Woodland Key Habitat.

regions were clearly separated in multivariate location in the NMDS (as indicated by almost all points grouped with points from the same region).

In summary, the results indicate that spruce-dominated Woodland Key Habitats have a higher variation in community composition (beta diversity) than commercially thinned spruce production stands. Between the forest types, there are no clear overall differences in community composition.

## Gamma diversity and feeding guilds

For the Jönköping region, most accumulation curves were similar (Fig 5). Regardless of species group, the reserve curves were generally lower than for the other forest types, but with wide and overlapping confidence intervals. For both saproxylic beetles (Fig 5A) and red-listed beetles (Fig 5B), the WKH curve was slightly higher than the thinned stand curve, but confidence intervals were overlapping. For wood consumers (Fig 5C) and fungivores (Fig 5D), the WKH and thinned stand curves were largely identical. For predators, the WKH curve was higher than the curves for thinned stands and reserves, with largely non-overlapping confidence intervals (Fig 5E), indicating higher gamma diversity in the WKHs.

For the Örebro region, accumulation curves were also mostly similar (Fig 6). For saproxylic beetles (Fig 6A) curves were virtually identical between the forest types. However, for red-listed beetles the WKH curve was higher (37.4 ± CI 7.6 species at highest extrapolation) than the unthinned and especially the thinned stand curves, with the confidence interval largely non-overlapping with the thinned stands (27.2 ± CI 6.7 species at highest extrapolation; Fig 6B), indicating higher gamma diversity in the WKHs. The curve for wood consumers was

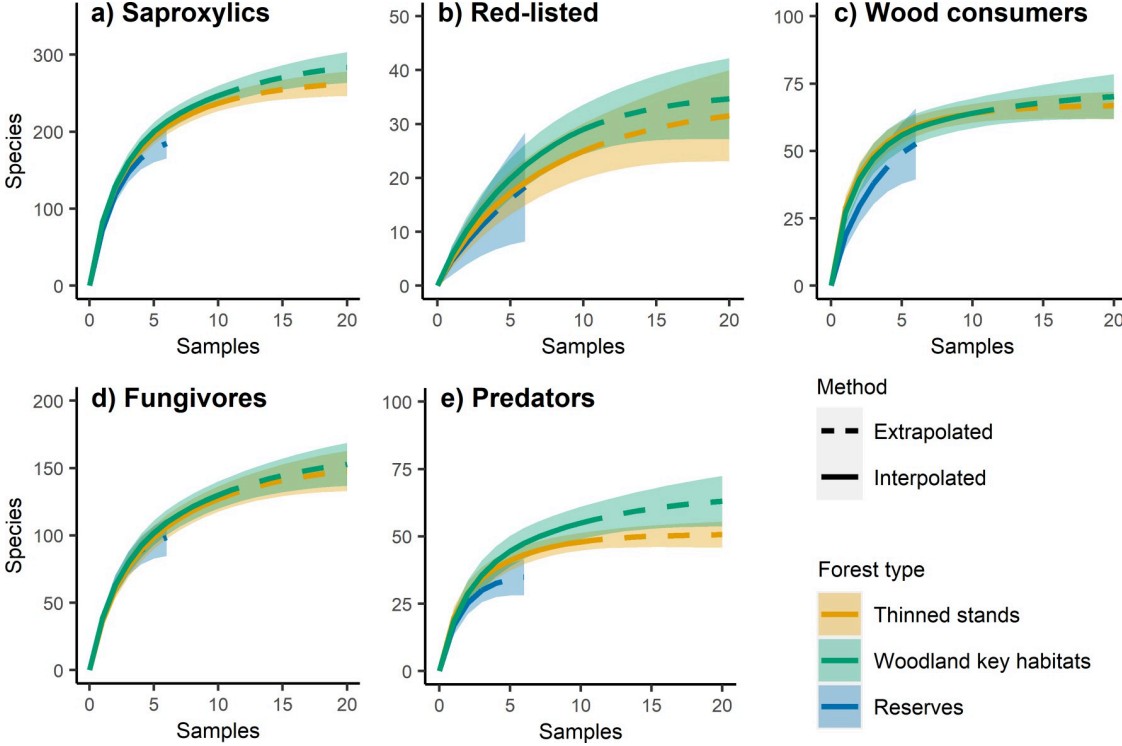

**Fig 5. Gamma diversity Jönköping.** a-e) Species accumulation curves with 95% confidence intervals for each forest type and species group from the Jönköping sample. Samples are forest stands (data from two traps per stand pooled), extrapolated to twice the original sample size.

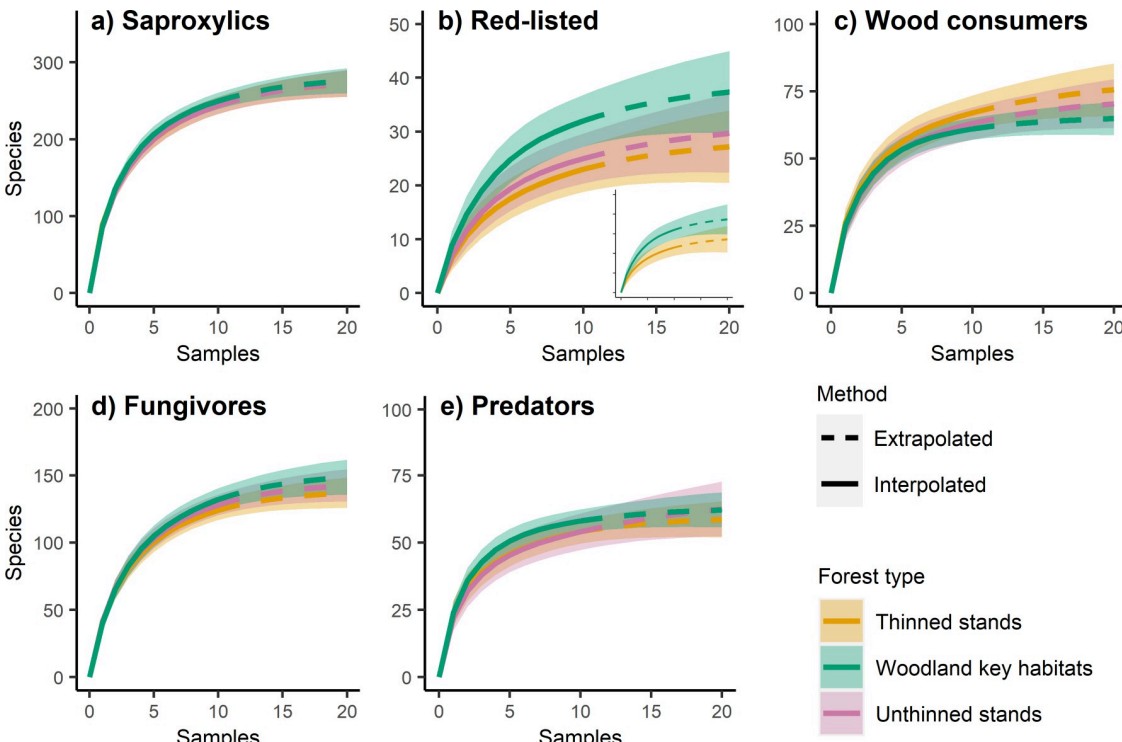

**Fig 6. Gamma diversity Örebro.** a-e) Species accumulation curves with 95% confidence intervals for each forest type and species group from the Örebro sample. Samples are forest stands (data from two traps per stand pooled), extrapolated to twice the original sample size. In b), the inlay shows the curves for the Woodland Key Habitats and the thinned stands, with the unthinned stands removed for clarity.

slightly higher for the thinned stands than for the unthinned stands, which in turn was higher than for the WKHs, although the confidence intervals for all three forest types were overlapping (Fig 6C). The curves for fungivores (Fig 6D) and predators (Fig 6E) were largely identical between the forest types.

In summary, the results indicate that for most of the species groups, total (gamma) diversity does not differ between the forest types. An exception is the red-listed species, for which gamma diversity is higher in spruce-dominated Woodland Key Habitats than in commercially thinned spruce production stands in one region (Örebro) but not the other (Jönköping).

## Discussion

For saproxylic beetles, I found lower alpha diversity in WKHs than in managed stands, but a higher beta and similar gamma diversity. For red-listed beetles, I found no statistically significant differences in alpha diversity, but gamma diversity was higher in WKHs in one region but not the other, possibly reflecting a biologically richer landscape in the former. These results highlight consequences of the choice of diversity measure and indicate the importance of spatial scale for the diversity of saproxylic beetles, especially red-listed ones.

### Alpha diversity and species composition

Prediction 1 predicted no difference in alpha diversity of saproxylic beetles between managed and unmanaged stands. Perhaps surprisingly, alpha diversity was higher in thinned stands than in WKHs. Clear differences in beetle diversity are often lacking between unmanaged

forests and production forest stages older than clear-cuts [77–81]. In a related study, we found no difference in alpha diversity between pre-commercially thinned spruce stands and WKHs [16]. These results mirror global trends, where hypothesized reductions in alpha diversity of various species groups as a consequence of anthropogenic influence on habitats are seldom found, or increases are seen instead [82–85].

Species dispersal has a positive (or unimodal) relationship to alpha diversity [31,86], and species associated with managed habitats should have little difficulty dispersing between patches in managed forest landscapes. Dispersal is likely more difficult for species associated with unmanaged forests, which are few, small and fragmented. Dispersal-limitation among species associated with unmanaged forests could mean that managed forests are more "fully-stocked" than unmanaged forests. If the regions used in this study consisted of mostly unmanaged forest, another pattern might be seen.

Contrary to prediction 2, I found no statistically significant differences in alpha diversity of red-listed beetles between the forest types, although the estimate for WKHs was higher than for the other types. This may be due to sampling: red-listed beetles are often rare, requiring very large samples to be properly assessed [62]. Furthermore, the diversity of red-listed saproxylic beetles may also need to be analyzed at larger scales than the local.

In contrast to the prediction of prediction 3, I found no clear differences in species composition among the forest types. It seems plausible that many species move freely between the forest types and do not view one or the other type as an impassable matrix [8], at least not in forests past the clear-cut stage [87]. Furthermore, nearby forest stands may not be the most appropriate scale to evaluate compositional differences, instead manifesting between e.g. regions with differing amounts of the forest types. If the amount of unmanaged, semi-natural forest (such as WKH) is much lower than that of managed forests, it is also possible that a potentially distinct fauna associated with unmanaged forests is strongly diluted by common immigrant species from surrounding managed stands [88,89]. In addition, associations of individual species to the different forest types may be obscured by a general community composition measure, and could be interesting to explore in more detail.

## Environmental factors

In contrast to prediction 4, I found no positive association of the environmental factors dead wood volume and dead wood diversity with saproxylic beetles, or red-listed beetles. Saproxylic beetle species are generally thought to follow the species-area relationship, with dead wood volume instead of area [90,91]. Thresholds of increasing saproxylic diversity at around 20 $m^3$/ha of dead wood in boreal forests have been proposed [92]. Given that this is well below what I found in the unmanaged stands (~82 $m^3$/ha on average in WKHs) and above what I found in the managed stands (~16 $m^3$/ha on average in thinned stands) it is perhaps surprising that I found no positive association with dead wood volume, and no difference in alpha diversity among the forest types. However, it should be noted that several of the WKH sites may have had inflated dead wood volume due to recent bark beetle attacks (pers. obs).

More important than dead wood volume per se may be dead wood diversity [37,93–95]. The span of dead wood diversity was quite limited in the current study, which may explain the absence of an association with saproxylic diversity. In a previous study, higher gamma diversity of certain saproxylic beetle groups in WKHs than in pre-commercially thinned spruce stands may have been partly explained by higher tree species diversity in the WKHs [16]. In this study, the WKHs were spruce-dominated, with quite similar dead wood diversity between managed and unmanaged stands. Roughly 60% of WKHs in southern and mid-Sweden are coniferous forests [96], dominated by Norway spruce or Scots pine. This spruce dominance in

Fennoscandian forests, managed and unmanaged, is largely a product of historical forest management [97].

Canopy openness had a slight negative influence on both saproxylic and red-listed beetles. This is contrary not just to prediction 4 but seems to contradict previous literature concluding that more saproxylic beetles prefer sun-exposed dead wood than shaded [38,98]. However, Norway spruce is late-successional, adapted to closed-canopy, small-scale gap-dynamics and old-growth stands that characterized much of pre-industrial Fennoscandia [99], and spruce-associated species likely have similar preferences.

### Beta and gamma diversity of saproxylic beetles

Consistent with prediction 5, I found higher beta diversity of saproxylic beetles in the WKHs than in the thinned stands. This suggests that largely the same "managed stand fauna" reoccurs, whilst the unmanaged stands are more heterogeneous. Beta diversity is negatively associated with dispersal, as it lets the same set of species populate many sites [31,86]. If, as reasoned above, the managed stand fauna exhibits higher dispersal, this could explain the lower beta diversity. Alternatively, the lower beta diversity in the managed stands could be due to lower among-stand diversity of forest structure and substrates (e.g. dead wood, see Fig 2).

In contrast to prediction 6, I did not find significantly higher gamma diversity of saproxylic beetles in the WKHs than in the other types. The definition of gamma or "regional" diversity is dependent on the scope of the study [31], and it is tightly linked to alpha and beta diversity [100]. In the scope of the present study, the higher alpha diversity in the thinned stands than in the WKHs may have compensated for the lower beta diversity. Given the difference in beta diversity between the forest types, a difference in gamma diversity would be expected if the scope of the study increased.

### Gamma diversity of red-listed beetles and feeding guilds

Consistent with prediction 7, there was a clear difference in the gamma diversity of red-listed beetles between managed and unmanaged stands in the region with a higher concentration of WKHs, but not in the region with lower concentration. This is in line with previous studies indicating the importance of the surrounding landscape for saproxylic beetle diversity, especially of red-listed beetles [26,101–103]. Given the paucity of semi-natural and unmanaged spruce forests in the Jönköping region, it is likely that many species dependent on these forest types are locally extinct. Although the Örebro region has a relatively higher concentration of WKHs than the Jönköping region, it still has a long history of anthropogenic influence [49]. Repeated in a more pristine region, it is possible the difference between the forest types would be even larger.

A hint at the increased importance of the larger spatial context for red-listed as compared to saproxylic beetles is also given by the random effect estimates in the GLMMs. While saproxylic beetles had variance associated with both sites and stands, red-listed beetles had considerable variance associated with sites, but virtually none with stands. This indicates that although the number of saproxylic beetles varied to some extent both among sites and between stands within sites, the red-listed beetles varied primarily at the larger, among-site scale.

As catches often vary between sampling years [104,105], and year is confounded with region in the present study, an alternate explanation of the contrast between regions/years is that WKHs functioned as better refuges for red-listed beetles during the hot and dry 2018 summer. Given climate change, this has potentially important implications for the coming value of unmanaged stands.

There were mixed results for the three feeding guilds, although largely not consistent with prediction 8. For the fungivores, which did not differ between the forest types, the prediction

presumed a substantially higher dead wood diversity and associated fungal diversity [106] in the WKHs, which seems not to have been the case. It is possible that the fine woody debris commonly left after thinning supports a relatively diverse and overlooked fungal diversity [107]. Wood consumers did indeed show higher gamma diversity in the thinned stands in the Örebro sample in accordance with the prediction, although the confidence intervals were partly overlapping. The pattern for predators was the reverse of the pattern for red-listed beetles, with higher gamma diversity in the WKHs in Jönköping but not in Örebro. Previous studies would suggest that predators should follow a similar pattern to red-listed beetles, being sensitive to forestry [108–110]. The pattern could either be caused by Jönköping WKHs being enriched in terms of predators, or thinned stands being impoverished. If the latter scenario is the case, a possible explanation could be that managed stands require nearby unmanaged stands in order to maintain a diverse fauna of predatory beetles. This might then be the case in Örebro, with a higher density of WKHs in the surrounding landscape, but not in Jönköping.

### Unthinned stands and reserves

Contrary to hypotheses 1, 2, 6 and 7, I did not find consistently lower alpha, beta or gamma diversity of saproxylic or red-listed beetles in the unthinned stands than in the other forest types. While thinning can clearly have a beneficial influence on saproxylic beetles living on tree species associated with more open conditions, such as oak [111,112], many species associated with spruce may instead prefer relatively shaded and undisturbed conditions, and self-thinning dead wood dynamics. This is congruent with results from northern Sweden, where [79] found similar alpha diversity of saproxylic beetles and red-listed saproxylic beetles in unthinned, commercially thinned and "unprotected mature" stands.

However, [79] found higher alpha diversity, and [90] both higher alpha and gamma diversity, in old-growth coniferous forests than in mature managed stands, contrasting with the results of the present study. Given the low sample size of reserves included in this study, no clear conclusion should be drawn. Furthermore, the differing results could once again be due to differing regional contexts of forest continuity or quality.

### Conclusions

This study helps clarify the importance of small, semi-natural unmanaged spruce stands for the conservation of forest biota, through a higher beta diversity of saproxylic beetles. Although managed stands seem to harbor many species, relative to other habitats, at the local scale, unmanaged stands are needed to counteract biotic homogenization.

The results also highlight the importance of considering scale, both for management and scientific studies of biodiversity. Gamma diversity of red-listed saproxylic beetles in WKHs was higher only in the region with a higher concentration of WKH stands. Without considering this regional context, a potentially important pattern would have been missed. The results indicate that for red-listed saproxylic beetles, the species most in need of conservation action, management should not ignore larger scales. Furthermore, as the contrasting results of alpha and beta diversity of saproxylic beetles in this study indicate, measuring only alpha diversity is insufficient. Studies comparing the diversity of managed and unmanaged forests only at the stand scale are likely to miss important context, resulting in simplified conclusions regarding the relative conservation value of forest types.

### Supporting information

**S1 Fig. Rank abundance curves.**
(PDF)

**S1 Table. PERMDISP based on Sørensen dissimilarity.**
(PDF)

## Acknowledgments

I thank Frank Götmark for much help with the study and manuscript, Sveaskog and Peter Bergman for providing study sites, and Olof Persson and Thomas Appelqvist for help with species identification.

## Author Contributions

**Conceptualization:** Oskar Gran.

**Data curation:** Oskar Gran.

**Formal analysis:** Oskar Gran.

**Funding acquisition:** Oskar Gran.

**Investigation:** Oskar Gran.

**Methodology:** Oskar Gran.

**Project administration:** Oskar Gran.

**Resources:** Oskar Gran.

**Software:** Oskar Gran.

**Visualization:** Oskar Gran.

**Writing – original draft:** Oskar Gran.

**Writing – review & editing:** Oskar Gran.

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
