## [Decision Letter · Decision Letter 0]

19 May 2022

PONE-D-22-04886Lower alpha, higher beta, and similar gamma diversity of saproxylic beetles in unmanaged compared to managed Norway spruce standsPLOS ONE

Dear Dr. Gran,

Thank you for submitting your manuscript to PLOS ONE. After careful consideration, we feel that it has merit but does not fully meet PLOS ONE’s publication criteria as it currently stands. Therefore, we invite you to submit a revised version of the manuscript that addresses the points raised during the review process.

We look forward to receiving your revised manuscript.

Kind regards,

Randeep Singh

Academic Editor

PLOS ONE

Journal Requirements:

Reviewers' comments:

Reviewer's Responses to Questions

**Comments to the Author**

1. Is the manuscript technically sound, and do the data support the conclusions?

Reviewer #1: Yes

Reviewer #2: No

Reviewer #3: Yes

Reviewer #4: Yes

2. Has the statistical analysis been performed appropriately and rigorously? 

Reviewer #1: Yes

Reviewer #2: No

Reviewer #3: Yes

Reviewer #4: I Don't Know

3. Have the authors made all data underlying the findings in their manuscript fully available?

Reviewer #1: Yes

Reviewer #2: Yes

Reviewer #3: Yes

Reviewer #4: Yes

4. Is the manuscript presented in an intelligible fashion and written in standard English?

Reviewer #1: Yes

Reviewer #2: No

Reviewer #3: Yes

Reviewer #4: Yes

5. Review Comments to the Author

Reviewer #1: Review of:

Lower alpha, higher beta, and similar gamma diversity of saproxylic beetles in unmanaged compared to managed Norway spruce stands

(PONE-D-22-04886)

General comment:

This is an important contribution dealing with forestry systems and the evaluation of biodiversity therein. Its value lies mainly in assessing a range of habitats produced by a boreal clear-cutting system, including young plantations (first study in project), thinning, harvesting stages, and set-asides for biodiversity (here, saproxylic beetles). The study is unusual and valuable in its approach, and in addition the use of classical diversity concepts in the analysis is well motivated.

The manuscript is well-written and the statistical analysis, as far as I can judge, is competent. See below for how the manuscript can, potentially, become even a bit better.

Minor comments:

Abstract: a good summary, but if possible add number of species and number of individuals (e.g. in parenthesis somewhere)

Row numbers:

48, maybe replace “important” with “species-rich and partly threatened”

60, change “decisions” to “recommendations”?

66, “…threatened with extinction”. Maybe omit ”with extinction”, and instead give the red-list categories used, in parenthesis (CR, EN, …).

76, “…but are often overlooked”. Better, perhaps: “… but are often overlooked in conservation research.”

101, change “looks at” to “examines”

109-110, “Unthinned stands, with less dead wood and canopy openness…” comes sudden and unexpected in wording, you may qualify with a parenthesis “(recall that “unthinned” refers to production stands)”, thus helping the reader

171-172, bedrock sentence may be deleted as long as you don’t present for each region separately (but of little importance anyway, drop?)

315, add “(Table 3)” at end of sentence to help readers.

329-330, add at sentence start (row 329), “Also for this group, canopy openness had a ….”

332, add Table 4, as for 315 (see above)

345, why is Gamma diversity presented second in Results, and not last? (now Beta diversity is last) Compare sequence of listing of hypotheses (Gamma last). And since Gamma diversity represents the largest spatial level, should it not be presented last in Results? Moreover, in title of manuscript the sequence is Alfa-Beta-Gamma.

376, should not “Community composition” have a separate sub-heading? NMDS and Fig 6 seem as related to Gamma as to Beta, since the it shows the whole communities (separated into two regions). For this, and comment 345, you may explain sequency / community composition in Materials and methods (e.g. last there, before Results).

376, as regards PERMDISP and Table 5, the results are clear, and pairwise testing has bearing on Beta diversity. You say in the Intro, “beta diversity represents the difference in species community among stands within a forest type”. Maybe it should be “between stands”, as “among stands” approaches Gamma diversity?

There might be other, complementary ways of illustrating Beta diversity. Could differences in species numbers, and/or differences in species abundances, be analysed by WKH nearest neighbor-distances (WKH site vs WKH site); thinned nearest neighbor-distance (thinned site vs thinned site), unthinned nearest neighbor-distance (unthinned site vs unthinned site)? Could such “between site” paired analysis be of value? (thus, comparing same stand type between sites). Model estimates (Table 5) are valuable, but do not illustrate absolute species/abundance Beta diversity patterns for the reader. This means excluding the real biological units (species, individuals) and showing more abstract statistical measures. (But Figs 2-5 are very good in showing “biology”.)

Discussion

409, change “there were” to “we found”

406-420, very good and interesting

423, change “could possibly” to “may well”

424-425, I suggest change “Furthermore, the main determinants of diversity of red-listed saproxylic beetles may manifest at larger scales than the local.” to “Furthermore, the diversity of red-listed saproxylic beetles must also be analyzed at larger scales than the local.

434, possibly add, “In addition, community composition is not the only test of hypothesis 3; individual species would be interesting to explore in more detail”

459, “…Norway spruce is late-successional, adapted to closed-canopy, small-scale gap-dynamics…”. But so is e.g. beech, with even stronger tolerance to closed canopies than Norway spruce. The contradiction of hypothesis 4 is interesting. What proportion of your species were “spruce species”? Please add, if possible.

469-470, You write “Alternatively, the lower beta diversity in the managed stands could be due to lower among-stand diversity of forest structure and substrates”. I assume this can be tested quantitatively with your data (WKH vs managed), though may be the subject of a different paper.

511-513, a bit hard to follow / understand. Also, were there more bark beetles in Jönköping, attracting predators?

519-520, “…many species associated with spruce…”. Again, what proportion of your species are spruce-connected?

527, drop “from this”

533, ”…seem to harbor many species at the local scale”,

change to “…seem to harbor many species, relative to other habitats, at the local scale”

533-534, consider omitting this part of sentence; “…encouraging for the potential of conservation-oriented management actions within these forests…”. Not necessary, and not clear.

541, consider changing “management might do best focusing on scales larger than the local”

to: “management should focus on both local and larger scales”

Reviewer #2: The artilcle written in thesis formate. Tiltle, abstract, introduction, results, discussion not according to Scinetific manners, it should be revised and resubmit again.

Introduction of subject experiment not well explained. Author should explain briefly why this study required to conduct? What are the study gaps you address? Results presentations not in scientific terms, it should need to revise profoundly. Future directions are also not presented well overloaded text should need to removed.

Reviewer #3: Review for PONE-D-22-04886

Overall, it is a worthwhile piece of work. It presents an important contribution to the scientific community concerning the saproxylic beetles’ diversity across forest types and spatial scales i.e. alpha, beta, and gamma diversity. The paper has no shortcomings, the study appears to be scientifically sound, the language clear, making it easy to follow.

My specific minor comments are outlined:

Lines 91-94: Amount of dead wood, diversity and canopy openness seem to be three different factors affecting saproxylic beetles’ diversity. Please revise “seems to be a major determinant” and “Both of these”.

Line 93: The authors maybe could provide a reference for species preferring shaded wood, too (e.g. Müller et al. 2015 - doi: 10.1111/1365-2664.12421 or other).

Line 154: The authors may specify for clarity that coordinates are in WGS84 coordinate reference system, using Decimal Degrees (DD) as its units. Also, the data source of background map could be added.

Line 156: Instead of “(Nilsson et al., 2019)” the authors could write “[27]”.

Table 1: For consistency, next to Canopy openness the authors should write “(%)” and remove the units from the numbers. Also, in Stand age units must be added i.e. years in order the table to stand on is own without reading the paper.

Line 197: Please also write the scientific name for Scots pine (Pinus sylvestris L.).

In Figures write the full terms, for example of WKH and in Tables explain all abbreviations eg.in Line 322 after “Confidence intervals” you can add CI in parentheses.

Reviewer #4: In this manuscript, the author compares alpha, beta, and gamma diversity of saproxylic beetles in managed versus unmanaged Norway spruce stands in central-southern Sweden. Author used field collected samples across two years from two different sampling region and found that alpha and beta diversity were higher for saproxylic beetles except for red-listed ones in thin stands and semi-natural thinned stands respectively whereas gamma diversity was higher for red-listed beetles in semi-natural thinned stands. The author further argues that all three (alpha, beta, and gamma) diversity measures need to be taken into consideration while carrying out conservation efforts. I really appreciate the amount of work the author has put into measuring plant diameters, and in collection and identification of thousands of beetles. Even though I like the manuscript, I have some suggestions which I feel will make the manuscript better:

My biggest reservation about this manuscript is the introduction section. I am still not convinced if all the hypothesis that the author lists are actual hypothesis as some of these feel like predictions to me. I wonder if it would be possible to narrow them down and a lot of the predictions can be brought into discussion sections when the author is discussing the findings. I also feel the introduction section can be more coherent. I appreciate the fact that the author has tried to put as much information as possible, however the information can be arranged properly such that it will be easier for the reader. For example, author introduces the study system in the first paragraph and do not come back to it until third paragraph and again till fifth or sixth paragraph.

Figure 2 and 3: I think these figures could be supplemental figures as they are not part of the result section of the manuscript.

I also think the background grid and color is making it difficult to read the figure. So, adding “theme_bw()” at the end of the ggplot2 code that the author already has can help give a clean background making it easier to comprehend the figure.

It would also be helpful to label the x-axis as “log (Diameter at breast height)” rather than mentioning it in the figure legend.

Result: All the result section is explained only in statistical terms. On its current form, the manuscript is inaccessible to a person who is not familiar with linear mixed models. So, it would be helpful to provide biological explanations on what those results mean in at least one or two simple sentences such that the manuscript would be accessible to a broader audience.

Line 30 : The sentence starting in “These results..” would make a better final sentence to the abstract.

Line 141, 144, and 145: Replace “sample” with “sampling area or sampling region”

Line 193: Add “,” after trap

Line 210: Please replace “;” with “,” and add “add” before dead trees

Line 211: Is it a common practice to estimate visually?

Line 237&238: Please rewrite the sentence as “environmental variables per forest type and region are summarized in Table 1”

6. PLOS authors have the option to publish the peer review history of their article (what does this mean?). If published, this will include your full peer review and any attached files.

Reviewer #1: No

Reviewer #2: **Yes: **Habib Ali

Reviewer #3: No

Reviewer #4: No

---

## [Author Response · Author response to Decision Letter 0]

21 Jun 2022

PLOS ONE

Journal Requirements:

Response: this information has been added (lines 252-255).

Reviewers' comments:

Reviewer's Responses to Questions

Comments to the Author

1. Is the manuscript technically sound, and do the data support the conclusions?

Reviewer #1: Yes

Reviewer #2: No

Reviewer #3: Yes

Reviewer #4: Yes

2. Has the statistical analysis been performed appropriately and rigorously?

Reviewer #1: Yes

Reviewer #2: No

Reviewer #3: Yes

Reviewer #4: I Don't Know

3. Have the authors made all data underlying the findings in their manuscript fully available?

Reviewer #1: Yes

Reviewer #2: Yes

Reviewer #3: Yes

Reviewer #4: Yes

4. Is the manuscript presented in an intelligible fashion and written in standard English?

Reviewer #1: Yes

Reviewer #2: No

Reviewer #3: Yes

Reviewer #4: Yes

5. Review Comments to the Author

Reviewer #1: Review of:

Lower alpha, higher beta, and similar gamma diversity of saproxylic beetles in unmanaged compared to managed Norway spruce stands

(PONE-D-22-04886)

General comment:

This is an important contribution dealing with forestry systems and the evaluation of biodiversity therein. Its value lies mainly in assessing a range of habitats produced by a boreal clear-cutting system, including young plantations (first study in project), thinning, harvesting stages, and set-asides for biodiversity (here, saproxylic beetles). The study is unusual and valuable in its approach, and in addition the use of classical diversity concepts in the analysis is well motivated.

The manuscript is well-written and the statistical analysis, as far as I can judge, is competent. See below for how the manuscript can, potentially, become even a bit better.

Response: I thank reviewer #1 for helpful and insightful comments on the manuscript.

Minor comments:

Abstract: a good summary, but if possible add number of species and number of individuals (e.g. in parenthesis somewhere)

Response: this has been added to lines 20-21.

Row numbers:

48, maybe replace “important” with “species-rich and partly threatened”

Response: the line has been changed to “a diverse species group of conservation concern”, as I believe this fits better in the sentence while still being in line with what reviewer #1 suggests.

60, change “decisions” to “recommendations”?

Response: line changed accordingly, and for same line, “incorrect” changed to “poor” as there is no such thing as the “correct” management.

66, “…threatened with extinction”. Maybe omit ”with extinction”, and instead give the red-list categories used, in parenthesis (CR, EN, …).

Response: line changed accordingly.

76, “…but are often overlooked”. Better, perhaps: “… but are often overlooked in conservation research.”

Response: line changed accordingly.

101, change “looks at” to “examines”

Response: line changed accordingly.

109-110, “Unthinned stands, with less dead wood and canopy openness…” comes sudden and unexpected in wording, you may qualify with a parenthesis “(recall that “unthinned” refers to production stands)”, thus helping the reader

Response: The lines have been changed, clarifying that both thinned stands and unthinned stands are production stands, which hopefully makes the wording less abrupt and easier for the reader to follow.

171-172, bedrock sentence may be deleted as long as you don’t present for each region separately (but of little importance anyway, drop?)

Response: the sentence has been omitted.

315, add “(Table 3)” at end of sentence to help readers.

Response: line changed accordingly.

329-330, add at sentence start (row 329), “Also for this group, canopy openness had a ….”

Response: line changed to “Also for red-listed species…” for clarity.

332, add Table 4, as for 315 (see above)

Response: line changed accordingly.

345, why is Gamma diversity presented second in Results, and not last? (now Beta diversity is last) Compare sequence of listing of hypotheses (Gamma last). And since Gamma diversity represents the largest spatial level, should it not be presented last in Results? Moreover, in title of manuscript the sequence is Alfa-Beta-Gamma.

Response: I have switched the order of the two sections. This also meant that the numbering for figures 4-6 changed, and the text has been updated to for this.

376, should not “Community composition” have a separate sub-heading? NMDS and Fig 6 seem as related to Gamma as to Beta, since the it shows the whole communities (separated into two regions). For this, and comment 345, you may explain sequency / community composition in Materials and methods (e.g. last there, before Results).

Response: The analyses of community composition and beta diversity are very closely related and complement each other, and both differences in location (i.e. composition) and dispersion (i.e. beta diversity) are visualized by the NMDS, so to me it makes much more sense to group these together. I have added some further explanation of the PERMDISP and NMDS in the methods section (lines 297, 299-301).

376, as regards PERMDISP and Table 5, the results are clear, and pairwise testing has bearing on Beta diversity. You say in the Intro, “beta diversity represents the difference in species community among stands within a forest type”. Maybe it should be “between stands”, as “among stands” approaches Gamma diversity?

Response: Beta diversity here is a measure of variation in community composition among multiple stands (in the NMDS, the average distance of a stand from the centroid of all points of the same forest type), so I think among is more fitting than between. But I have rephrased the definition in lines 79-80 to make it more straightforward.

There might be other, complementary ways of illustrating Beta diversity. Could differences in species numbers, and/or differences in species abundances, be analysed by WKH nearest neighbor-distances (WKH site vs WKH site); thinned nearest neighbor-distance (thinned site vs thinned site), unthinned nearest neighbor-distance (unthinned site vs unthinned site)? Could such “between site” paired analysis be of value? (thus, comparing same stand type between sites). Model estimates (Table 5) are valuable, but do not illustrate absolute species/abundance Beta diversity patterns for the reader. This means excluding the real biological units (species, individuals) and showing more abstract statistical measures. (But Figs 2-5 are very good in showing “biology”.)

Response: See Anderson et al. 2006 [75] for problems associated both with using dissimilarities between pairs of sampling units (p. 684 “A test for…”), and using raw abundances, i.e. “real biological units” (p. 685 “Potential pitfalls…”) when testing differences in beta diversity between groups of sampling units (e.g. stands of different forest types). I follow the recommendations from Anderson et al. 2011 [74] on the appropriate way to test differences in multivariate variation (in this case beta diversity) between levels of a categorical factor (in this case forest type).

Discussion

409, change “there were” to “we found”

Response: text changed accordingly

423, change “could possibly” to “may well”

Response: I changed “could possibly” to “may”

424-425, I suggest change “Furthermore, the main determinants of diversity of red-listed saproxylic beetles may manifest at larger scales than the local.” to “Furthermore, the diversity of red-listed saproxylic beetles must also be analyzed at larger scales than the local.

Response: line changed to “Furthermore, the diversity of red-listed saproxylic beetles may also need to be analyzed at larger scales than the local.”

434, possibly add, “In addition, community composition is not the only test of hypothesis 3; individual species would be interesting to explore in more detail”

Response: added “In addition, associations of individual species to the different forest types may be obscured by a general community composition measure, and could be interesting to explore in more detail.”

459, “…Norway spruce is late-successional, adapted to closed-canopy, small-scale gap-dynamics…”. But so is e.g. beech, with even stronger tolerance to closed canopies than Norway spruce. The contradiction of hypothesis 4 is interesting. What proportion of your species were “spruce species”? Please add, if possible.

Response: I have added a section on host tree associations at the start of the results section, lines 309-314.

469-470, You write “Alternatively, the lower beta diversity in the managed stands could be due to lower among-stand diversity of forest structure and substrates”. I assume this can be tested quantitatively with your data (WKH vs managed), though may be the subject of a different paper.

Response: In theory this could be tested, but I believe this to be outside the scope of the present paper, and would perhaps require additional measurements of stand characteristics.

511-513, a bit hard to follow / understand. Also, were there more bark beetles in Jönköping, attracting predators?

Response: I have rephrased the section to make it easier to follow. There were indeed slightly more wood consumers in the Jönköping sample, although this general pattern would not explain the difference between the forest types.

519-520, “…many species associated with spruce…”. Again, what proportion of your species are spruce-connected?

Response: See above.

527, drop “from this”

Response: text change accordingly

533, ”…seem to harbor many species at the local scale”,

change to “…seem to harbor many species, relative to other habitats, at the local scale”

Response: text change accordingly

533-534, consider omitting this part of sentence; “…encouraging for the potential of conservation-oriented management actions within these forests…”. Not necessary, and not clear.

Response: text change accordingly

541, consider changing “management might do best focusing on scales larger than the local”

to: “management should focus on both local and larger scales”

Response: text changed to “management should not ignore larger scales”

Reviewer #2: 

Response: I thank reviewer #2 for comments on the manuscript. However, I find some of the comments difficult to follow as they do not refer to specifics, and at times refer to sections that are not present in the manuscript (e.g. no experiment was carried out; the article does not contain a section on future directions).

The artilcle written in thesis formate. Tiltle, abstract, introduction, results, discussion not according to Scinetific manners, it should be revised and resubmit again.

Response: I believe the manuscript follows the structure set out in the PLoS ONE submission guidelines.

Introduction of subject experiment not well explained. 

Response: I am unclear as to what reviewer #2 is referring to here. There was no experiment carried out in this study. 

Author should explain briefly why this study required to conduct? What are the study gaps you address? 

Response: This is addressed several times in the introduction, e.g. lines 45-47, 61-65, 81-83.

Results presentations not in scientific terms, it should need to revise profoundly. 

Response: It is hard to address this point without specific examples. I believe the results section uses scientific terminology throughout. The comment also contrasts with comments from reviewer #4 indicating the opposite, i.e. that the results section should use less technical language.

Future directions are also not presented well overloaded text should need to removed.

Response: I am unsure as to what part of the text this comment is concerning, as the manuscript does not contain a "Future directions" section.

Reviewer #3: Review for PONE-D-22-04886

Overall, it is a worthwhile piece of work. It presents an important contribution to the scientific community concerning the saproxylic beetles’ diversity across forest types and spatial scales i.e. alpha, beta, and gamma diversity. The paper has no shortcomings, the study appears to be scientifically sound, the language clear, making it easy to follow.

Response: I thank reviewer #3 for helpful and insightful comments on the manuscript.

My specific minor comments are outlined:

Lines 91-94: Amount of dead wood, diversity and canopy openness seem to be three different factors affecting saproxylic beetles’ diversity. Please revise “seems to be a major determinant” and “Both of these”.

Response: the lines have been changed accordingly.

Line 93: The authors maybe could provide a reference for species preferring shaded wood, too (e.g. Müller et al. 2015 - doi: 10.1111/1365-2664.12421 or other).

Response: the reference has been added.

Line 154: The authors may specify for clarity that coordinates are in WGS84 coordinate reference system, using Decimal Degrees (DD) as its units. Also, the data source of background map could be added.

Response: the figure caption has been changed accordingly, and the source of the background map added.

Line 156: Instead of “(Nilsson et al., 2019)” the authors could write “[27]”.

Response: the reference has been corrected accordingly

Table 1: For consistency, next to Canopy openness the authors should write “(%)” and remove the units from the numbers. Also, in Stand age units must be added i.e. years in order the table to stand on is own without reading the paper.

Response: the table has been changed accordingly

Line 197: Please also write the scientific name for Scots pine (Pinus sylvestris L.).

Response: I have tried to be consistent regarding scientific names, and as Scots pine is mentioned earlier with scientific name in line 161, I have not provided the scientific name here.

In Figures write the full terms, for example of WKH and in Tables explain all abbreviations eg.in Line 322 after “Confidence intervals” you can add CI in parentheses.

Response: I have added explanations for all abbreviations in the table and figure legends, although I have retained the abbreviation WKH in most of the figures as Woodland Key Habitats is too long to fit and would require reducing the text size.

Reviewer #4: In this manuscript, the author compares alpha, beta, and gamma diversity of saproxylic beetles in managed versus unmanaged Norway spruce stands in central-southern Sweden. Author used field collected samples across two years from two different sampling region and found that alpha and beta diversity were higher for saproxylic beetles except for red-listed ones in thin stands and semi-natural thinned stands respectively whereas gamma diversity was higher for red-listed beetles in semi-natural thinned stands. The author further argues that all three (alpha, beta, and gamma) diversity measures need to be taken into consideration while carrying out conservation efforts. I really appreciate the amount of work the author has put into measuring plant diameters, and in collection and identification of thousands of beetles. Even though I like the manuscript, I have some suggestions which I feel will make the manuscript better:

Response: I thank reviewer #4 for helpful and insightful comments on the manuscript.

My biggest reservation about this manuscript is the introduction section. I am still not convinced if all the hypothesis that the author lists are actual hypothesis as some of these feel like predictions to me. I wonder if it would be possible to narrow them down and a lot of the predictions can be brought into discussion sections when the author is discussing the findings. 

Response: I accept reviewer #4’s point that they are perhaps more appropriately described as predictions than hypotheses, and have modified the text accordingly. I nevertheless believe they are supported by the preceding introduction, and I think they provide a useful structure to the paper. Having them present from the start should make it easier for the reader to follow the different lines of inquiry present in the paper, and provides a clear connection between the introduction and discussion sections. This being said, if the editor still believes that some of these should be combined, simplified or removed from the introduction, I will not object.

I also feel the introduction section can be more coherent. I appreciate the fact that the author has tried to put as much information as possible, however the information can be arranged properly such that it will be easier for the reader. For example, author introduces the study system in the first paragraph and do not come back to it until third paragraph and again till fifth or sixth paragraph.

Response: The section explaining the theoretical framework of alpha-beta-gamma diversity may be somewhat incongruous with the rest of the introduction, although it is important establishing knowledge. I have moved the paragraph to a later part of the introduction (lines 78-91), in order to hopefully facilitate better flow and coherency.

Figure 2 and 3: I think these figures could be supplemental figures as they are not part of the result section of the manuscript.

Response: While they are not part of the results section, I believe they nonetheless are valuable in characterizing the stands, and have direct bearing on the interpretation of the results. Reviewer #1 also highlights these as good, in showing concrete “biology”. Consequently, I have left them in the main text, but ultimately I leave it up to the discretion of the editor whether they should be moved to the supplementary material.

I also think the background grid and color is making it difficult to read the figure. So, adding “theme_bw()” at the end of the ggplot2 code that the author already has can help give a clean background making it easier to comprehend the figure.

Response: While removing the background will possibly give a cleaner look, it will make it more difficult to identify which series of points are associated with which tree species, especially should the paper be viewed in black and white (e.g. printed). Additionally, for figure 3, leaving out the background will make it harder to identify the discrete facets (i.e. dead wood type x forest type combination). As such, I have chosen to retain the background.

It would also be helpful to label the x-axis as “log (Diameter at breast height)” rather than mentioning it in the figure legend.

Response: Renaming the x-axis as log(Diameter…) would not be correct, as this would indicate that the values have been log transformed. Instead, the raw values have been plotted on a logarithmic scale.

Result: All the result section is explained only in statistical terms. On its current form, the manuscript is inaccessible to a person who is not familiar with linear mixed models. So, it would be helpful to provide biological explanations on what those results mean in at least one or two simple sentences such that the manuscript would be accessible to a broader audience.

Response: To facilitate understanding, I have added a short summary of the main results to the end of each results section: lines 362-366, 391-394, 427-431. 

Line 30 : The sentence starting in “These results..” would make a better final sentence to the abstract.

Response: abstract changed accordingly

Line 141, 144, and 145: Replace “sample” with “sampling area or sampling region”

Response: text changed accordingly

Line 193: Add “,” after trap

Response: text changed accordingly

Line 210: Please replace “;” with “,” and add “add” before dead trees

Response: text changed accordingly

Line 211: Is it a common practice to estimate visually?

Response: Ocular estimation is used in forestry, e.g. for standing deadwood in The Swedish National Forest Inventory (see p. 217 of handbook [in Swedish] https://www.slu.se/globalassets/ew/org/centrb/rt/dokument/faltinst/ris_faltinstruktion_2011_hela.pdf). Although physically measuring the top diameter would obviously have been preferable, this was not practically possible in this case. The method is unlikely to have had much influence on the end result, as the difference in diameter between the bottom and tops of snags was usually minor, and the number of snags where this method was employed was small.

Line 237&238: Please rewrite the sentence as “environmental variables per forest type and region are summarized in Table 1”

Response: Response: text changed accordingly

---

## [Editor Report · Decision Letter 1]

24 Jun 2022

Lower alpha, higher beta, and similar gamma diversity of saproxylic beetles in unmanaged compared to managed Norway spruce stands

PONE-D-22-04886R1

Dear Dr. Gran,

We’re pleased to inform you that your manuscript has been judged scientifically suitable for publication and will be formally accepted for publication once it meets all outstanding technical requirements.

Kind regards,

Randeep Singh

Academic Editor

PLOS ONE
---

## [Editor Report · Acceptance letter]

30 Jun 2022

PONE-D-22-04886R1 

Lower alpha, higher beta, and similar gamma diversity of saproxylic beetles in unmanaged compared to managed Norway spruce stands 

Dear Dr. Gran:

I'm pleased to inform you that your manuscript has been deemed suitable for publication in PLOS ONE. Congratulations! Your manuscript is now with our production department. 

Kind regards, 

on behalf of

Dr. Randeep Singh 

Academic Editor

PLOS ONE